# Energy implications of the 21st century agrarian transition

Lorenzo Rosa [1,2], Maria Cristina Rulli [3], Saleem Ali[4,5✉], Davide Danilo Chiarelli [3], Jampel Dell'Angelo[6], Nathaniel D. Mueller[7,8], Arnim Scheidel [9], Giuseppina Siciliano[10] & Paolo D'Odorico [1]

The ongoing agrarian transition from small-holder farming to large-scale commercial agriculture is reshaping systems of production and human well-being in many regions. A fundamental part of this global transition is manifested in large-scale land acquisitions (LSLAs) by agribusinesses. Its energy implications, however, remain poorly understood. Here, we assess the multi-dimensional changes in fossil-fuel-based energy demand resulting from this agrarian transition. We focus on LSLAs by comparing two scenarios of low-input and high-input agricultural practices, exemplifying systems of production in place before and after the agrarian transition. A shift to high-input crop production requires industrial fertilizer application, mechanization of farming practices and irrigation, which increases by ~5 times fossil-fuel-based energy consumption compared to low-input agriculture. Given the high energy and carbon footprints of LSLAs and concerns over local energy access, our analysis highlights the need for an approach that prioritizes local resource access and incorporates energy-intensity analyses in land use governance.

[1] Department of Environmental Science, Policy, and Management, University of California, Berkeley, Berkeley, CA, USA. [2] Institute of Energy and Process Engineering, ETH Zurich, 8092 Zurich, Switzerland. [3] Department of Civil and Environmental Engineering, Politecnico di Milano, Milan, Italy. [4] Department of Geography and Spatial Sciences, University of Delaware, Newark, DE, USA. [5] Sustainable Minerals Institute, University of Queensland, St Lucia, Australia. [6] Institute for Environmental Studies (IVM), Vrije Univeristeit Amsterdam, Amsterdam, The Netherlands. [7] Department of Ecosystem Science and Sustainability, Colorado State University, Fort Collins, CO, USA. [8] Department of Soil and Crop Sciences, Colorado State University, Fort Collins, CO, USA. [9] Institut de Ciència i Tecnologia Ambientals (ICTA-UAB), Universitat Autònoma de Barcelona, Bellaterra, Spain. [10] Centre for Development, Environment and Policy, SOAS, University of London, London, UK. ✉email: saleem@alum.mit.edu

The historical limits of traditional agriculture, namely, soil fertility, water, and energy have been overcome by the process of industrialization implemented through the green revolution[1]. The adoption of fossil-fuel-based industrial fertilizers, new cultivars, and machineries—including motorized pumps—permitted to remove fundamental limitations associated with soil nutrients, water resources, and labor. Advances in the nitrogen fertilizer industry, irrigation, and other technologies increased land efficiency in agriculture[2], but did not necessarily lead to inputs' savings[3–5]. As a result, global crop production has tripled in the last 50 years, causing strong environmental impacts[3].

A crucial challenge for humanity is to sustainably meet future food demand[6], while limiting agriculture's environmental footprint[7]. Assessments on how to reduce the environmental footprint of agriculture commonly point to a combination of measures, including the promotion of dietary shifts, reduction of food waste, and halting agricultural expansion by closing yield gaps—the difference between maximum attainable yield and current yield for specific crops—through agricultural intensification[8].

Increasing yields do not guarantee local hunger alleviation or reduced pressure on local natural resources[3], though evidence supports benefits to global health[9] and land use[2]. Agricultural intensification rarely produces win-win situations[10]; only a few cases have demonstrated the contribution of agricultural intensification to multiple Sustainable Development Goals (SDGs), such as ending hunger (SDG2) and sustainable land-use (SDG15)[10]. Controversies over the role of agricultural intensification for land use and users have intensified with the recent unprecedented rise in large-scale land acquisitions (LSLAs) globally[11,12]. LSLAs refer to long-term and large-scale acquisitions of land property or use rights through domestic and foreign actors, which have been sparked by—among other factors—food security concerns and the rediscovery of agriculture as a key investment sector following the 2008 food, financial, and energy crises[13]. According to the Land Matrix—a joint international initiative collecting data on LSLAs since 2000—90 million hectares of land (about the surface area of Venezuela) have been acquired globally by investors since 2000[14].

A few studies have discussed the potential opportunities of LSLAs for agricultural development, for instance, their potential to increase yields[13] and economic benefits for the country through large-scale investments into farmland[15]. However, other studies have described these land deals as land grabs that entail a vast range of adverse impacts on local users, including land dispossession and livelihood loss[16–18]. There is strong evidence that large-scale investments in agriculture by agribusiness corporations have been implemented at the cost of smallholders, traditional users, Indigenous people, and more vulnerable segments of the rural population[12,16]. LSLAs could paradoxically increase crop production, while undermining local food security through the production of energy-rich but nutrient-poor crops for export markets[14]. While this radical transformation of agrarian systems through LSLAs has been extensively investigated in relation to its political implications[19,20] and its impacts on property systems[21], rural livelihoods[22], crop yields[14], water use and redistribution[23], food security[24–26], environmental impacts[27], and carbon emissions[28], however, with few exceptions[29], its energy implications remain poorly understood.

Global food systems are major energy users and contributors to climate change. Food systems consume 15–30% of global primary energy[30] and emit 25–34% of global total greenhouse gas (GHG) emissions[31–33]. Although 35% of these emissions are caused by dairy and meat production, the remaining share is from activities pertaining to crop production for direct human consumption[31]. While, GHG emissions associated with deforestation and land use change over 40 million hectares of LSLAs have been recently estimated to 8 gigatons $CO_2$-equivalent in the 2000–2016 period[28], the energy and related GHG emissions associated with the agrarian transition induced by LSLAs and other dynamics favoring the expansion of commercial farming have often been overlooked.

Here, we evaluate the energy and fossil-fuel implications of the agricultural transition that is being globally promoted by LSLAs. In particular, we focus on the energy-related systematic changes that happen when moving from traditional low-input, labor-intensive agricultural systems to intensified systems of crop production with high-input of fertilizers, pesticides, machineries, mechanized irrigation, and comparatively less labor associated with the transition. Using a variety of data[34–36], we estimate pre- and post-LSLA energy inputs in crop production over acquired lands. In particular, we assess the energy intensity that LSLAs would have in the case a low-input or a high-input farming were developed. We also calculate the energy intensity of irrigation over land deals considering different irrigation technologies. Finally, we discuss strategies to reduce the energy intensity of farming, decrease reliance on fossil-fuel-based technologies, and provide policy recommendations that might help to lower the carbon emissions. For our analysis, we consider 197 land deals with size greater than 200 hectares, obtained from the Land Matrix dataset[36] (Fig. 1).

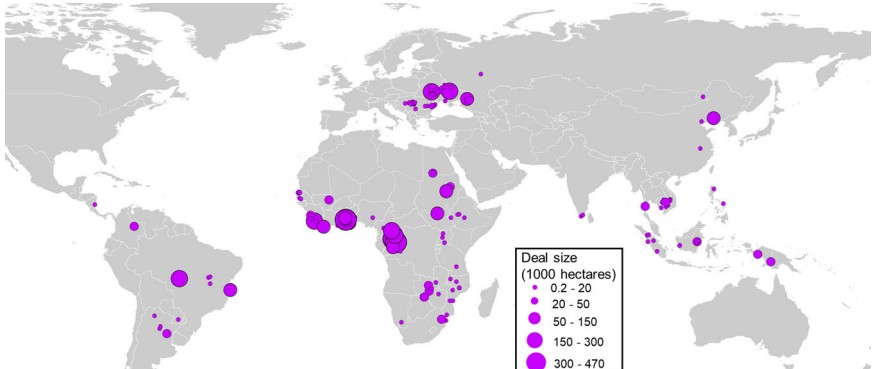

**Fig. 1 Geographical distribution of large-scale land acquisitions (LSLAs) considered in this study.** We consider 197 land deals for agricultural use for which the geographic coordinates were available in the Land Matrix database[36]. These land deals are located in 39 countries and account for 4.07 million hectares of acquired land across Africa (73 deals, 2.4 Mha), Asia (43 deals and 0.58 Mha), Europe (33 deals and 0.54 Mha), and Latin America (11 deals and 0.55 Mha).

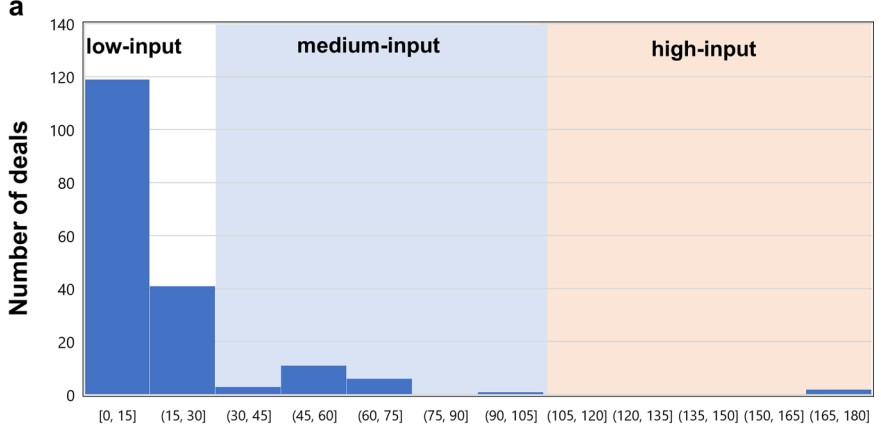

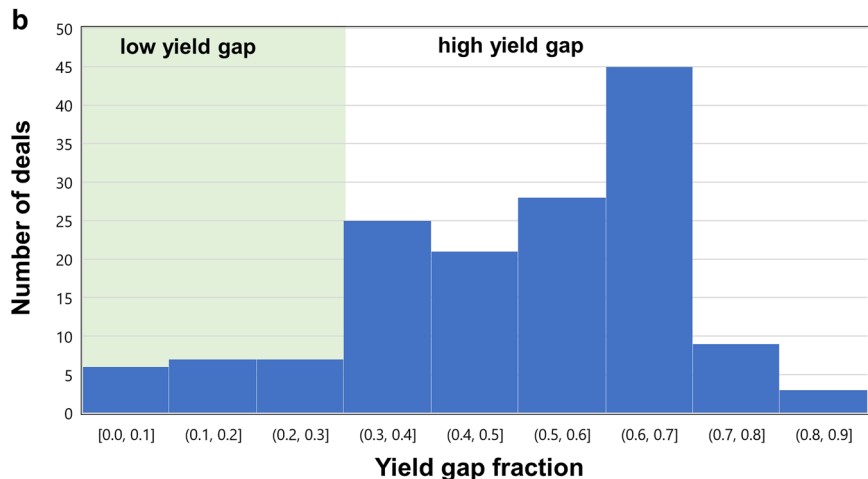

**Fig. 2 Nitrogen application rates and yield gap closure levels before large-scale land acquisition (LSLA). a** Shows average synthetic nitrogen application per harvested hectare in each deal (kg N per ha). An application smaller than 30 kg ha$^{-1}$ is considered low-input agriculture; between 30 and 100 kg ha$^{-1}$ medium-input agriculture; and greater than 100 kg ha$^{-1}$ fairly high-input agriculture. **b** Agricultural productivity levels measured in terms of yield-gap fraction of major crops before LSLAs. Yield-gap fractions lower than 0.3 show land deals involving lands with high agricultural productivity. Data source: Mueller et al.[35].

The energy implications of LSLAs have not yet been investigated in depth. Given that many existing policy mechanisms support LSLAs and other pathways to agricultural intensification[37], this study provides a first examination of the fossil energy and associated energy implications of agricultural transitions following LSLAs.

## Results

**Agricultural productivity before LSLAs**. To ascertain whether the development of commercial crop production intended by the land deals would entail agricultural intensification, we considered fertilizer application rates in areas prior to land acquisition, as well as the fertilizer applications needed to reach maximum attainable yields[35]. Figure 2a shows that 80% of the deals involve land that was previously characterized by low-input agriculture. Only 1% of the deals had high-input application of synthetic fertilizers before LSLAs. Figure 2b shows that 85% of the deals occurred on land with low agricultural productivity and high yield gaps. This implies that, for the purpose of developing large-scale commercial crop production, additional inputs, such as fertilizers and water, are needed for most of the land acquired through large-scale land investments.

**Energy usage of low- and high-input agriculture**. Figure 3 shows the energy intensity per area for the main crops intended by LSLAs at the farm level under low- and high-input agriculture scenarios. Fossil-fuel-based energy inputs are from labor, machinery, fertilizers, chemicals, fuels, and seeds. The figure also shows the energy intensity of oil palm and jatropha mills for biodiesel production. Among the four main staple crops (rice, wheat, maize, and soybean, which account for 70% of global food production), high-input rice production is the most energy-intensive agricultural practice. Because soybean and pulses have the ability to fix nitrogen directly from the atmosphere—requiring lower amounts of nitrogen fertilizers—their energy intensity is relatively low compared to other staple crops. Interestingly, low-input agriculture with sorghum—a widespread crop in Sub-Saharan Africa—is the least energy-intensive agricultural practice. We find that a transition from sorghum and millet to cash crops or staple crops might have profound energy implications. Cotton has the highest energy-intensity even for low-input options, which also raises important questions about non-food crop choice for land use as well as broader consumer options for end-use materials. Groundnut crops show the largest variation in energy intensity for low- and high-input cultivation forms.

We further analyzed the aggregate area of intended crops to estimate fossil-fuel energy requirements and related carbon

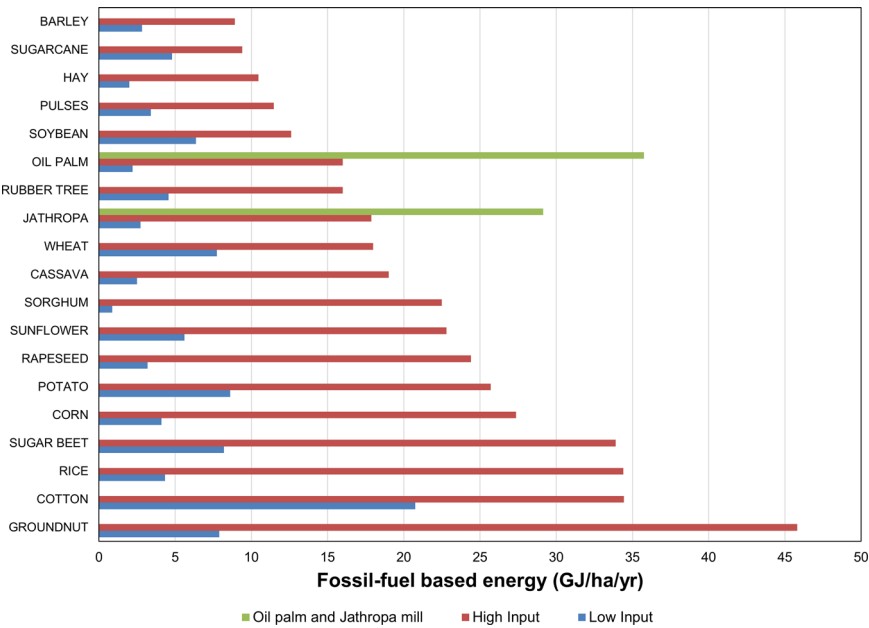

**Fig. 3 Fossil-fuel-based energy intensity at the farm level of low- and high-input agriculture.** Fossil-fuel-based energy inputs are from labor, machinery, fertilizers, chemicals, fuels, and seeds. See Supplementary Table 1 for data sources.

emissions in the context of global land-use change triggered by LSLAs (Fig. 4). Oil palm is the most frequently proposed crop of LSLAs, accounting for 25% (1.02 Mha) of the total area of land deals considered here (Fig. 4a). This raises concerns about high levels of GHG emissions from palm oil plantations, particularly when they are associated with forest conversion[38], as well as about the high fossil energy requirements in palm oil milling for biodiesel production (see Fig. 3). Other important crops intended by LSLAs are other tree plantations, followed by a range of food and fodder crops (Fig. 4a).

We assumed that the whole deal area cultivated in low-input agriculture transitioned to high-input agriculture. Developing the 4.07 Mha of intended crops under a scenario of low-input agriculture would require an annual amount of 3 million barrels of oil equivalent as energy input (Fig. 4b). However, given that many land deals seek large-scale commercial crop production, high-input, capital-intensive production can be expected to prevail in many land deals[28,39]. Cultivating the intended crops with high-input agriculture would require ~5 times more fossil-fuel-based energy (or 15 million barrels of oil equivalent per year).

We assessed the additional GHG emissions that would be generated through high-input farming over LSLAs. Assuming that modern agricultural technologies are all powered with oil and considering an average GHG emission intensity of oil equal to 492.6 kg $CO_2$ equivalent per barrel of oil (see "Methods"), we find that high-input farming over LSLAs would emit an additional 6 million tons of $CO_2$ per year (approximately the emission of Uganda or Uruguay in the year 2017).

Under a business-as-usual strategy, large-scale farms are expected to be established on the entire transacted area and are expected to be cultivated and used for intensified crop production[28] with high-inputs to maximize agricultural productivity. This implementation strategy yields the highest potential energy use and provides an upper-bound estimate. Similarly, an upper-bound GHG emissions' estimate is provided assuming that additional energy demand is met entirely with fossil-fuel-based energy sources. Different pathways of development can fuel high-input agriculture through less carbon-intensive energy sources and therefore reduce carbon emissions. Inclusion of these factors,

as well as lower attainable yields, and agroecological practices such as crop diversification, natural fertilizer use, biological pest control, deficit irrigation, and other soft-path water-management practices are expected to reduce the amount of fertilizers and irrigation water, and therefore reduce energy usage. We tested the sensitivity of our results by assuming that 75% of the transacted deal area is cultivated, and considering an 80% attainable yield. Moreover, considering global primary energy demand statistics, we assumed that 80% of energy inputs are powered by fossil fuels and the remaining fraction by renewable energy[40]. This conservative scenario would lower GHG emissions of high-input farming by 3 million tons of $CO_2$ per year, or half of the GHG emissions of a business-as-usual strategy.

**Energy usage for irrigation on LSLAs.** In addition to nutrients, water is another important input necessary to increase yields and avoid crop growth under water stress[41]. The use of irrigation enables reliable water supply while boosting crop productivity[42]. However, in many geographical settings, irrigation practices are unsustainable because their water requirements exceed local renewable water availability and irrigation induces the depletion of environmental flows and groundwater stocks[43]. Using recent irrigation sustainability assessments[44], we find that 20% of land deals were located in regions where water resources are insufficient to sustainably meet irrigation demand.

Meeting crop water requirements has further energy implications[45]. Irrigation requires energy to transfer water from the withdrawal source to the field—unless the local topography allows for gravity irrigation. We use the WATNEEDS crop water model[46] to assess the energy intensity of irrigation (i.e., the irrigation energy demand per unit of area) of each land deal, considering two of the most widespread irrigation systems: sprinkler and surface irrigation (Fig. 5). Because sprinkler irrigation has a higher operating pressure than surface irrigation, our results show that sprinkler irrigation over LSLAs requires more energy than surface irrigation—though surface irrigation requires larger volumes of water because of its lower efficiency (Fig. 5). Furthermore, the combination of crop type and climate in Eastern Europe, Asia, and Latin America is such

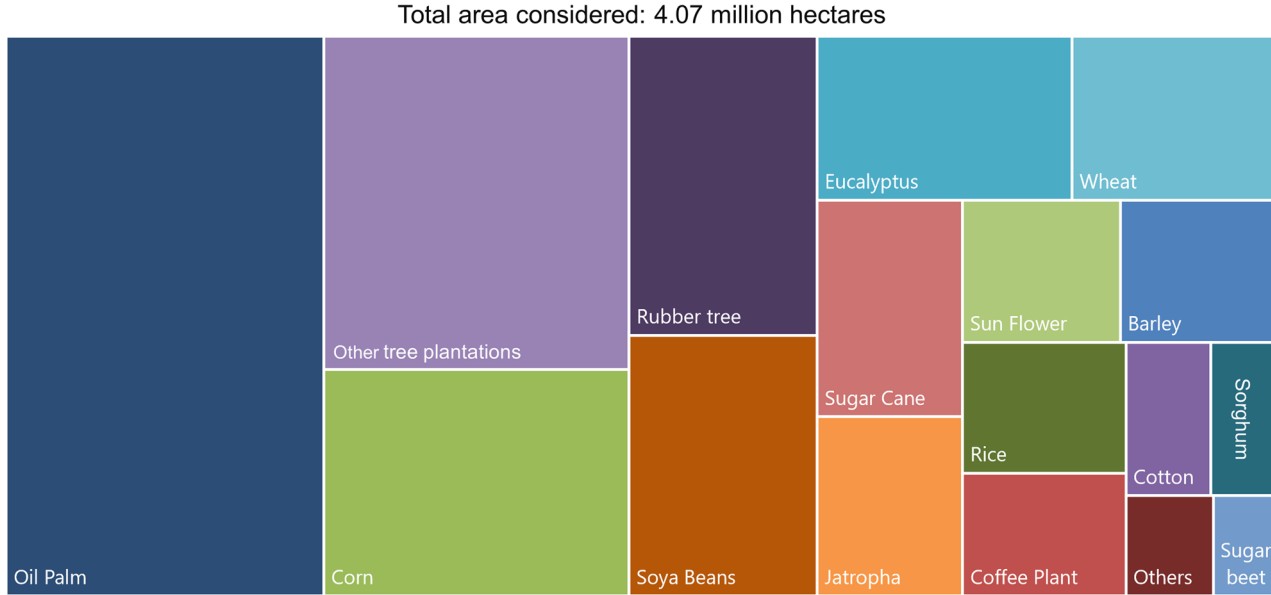

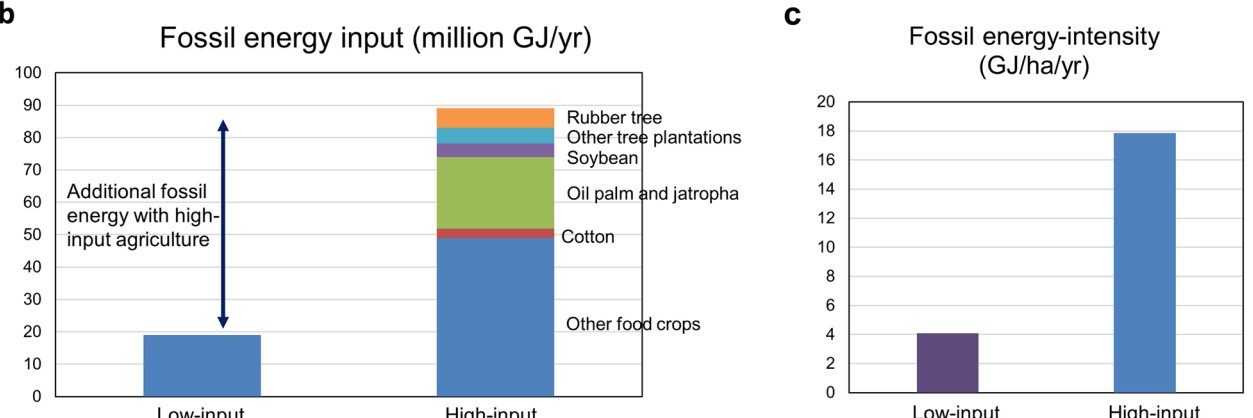

**Fig. 4 Intended crops for agricultural use over large-scale land acquisitions (LSLAs) and their aggregate fossil energy footprint under low- and high-input agriculture scenarios. a** Area (Mha) of intended crops for the sample of 197 land deals intended for agricultural use with size greater than 200 hectares, obtained from the Land Matrix dataset[36]. **b** Aggregated fossil-fuel-based energy input (million GJ yr$^{-1}$) for the intended crop area under low- and high-input agricultural scenarios. **c** Aggregated fossil energy intensity of low- and high-input agriculture over LSLAs. Energy inputs from oil palm and jatropha milling were not accounted for in this assessment. Source: Estimates based on data from Supplementary Table 1 and Land Matrix[36]). Note: one barrel of oil equivalent is equal to 6.1 GJ.

that land acquisitions display lower irrigation water requirements and therefore smaller specific energy needs (Fig. 5a). Conversely, Africa has the highest energy requirement for irrigation purposes, due to the combination of dry climate conditions and cultivated crops with high water needs (including oil palm and sugarcane). Because of the perennial features of these biofuel crops, results also show that land deals for biofuel are in general the most energy-intensive land uses in relation to the energy required for irrigation (Fig. 5b). We also show that if acquired lands were irrigated with sprinkler irrigation, an additional 4.3 million barrels of oil equivalent per year (or 26 million GJ) of fossil-fuel-based energy compared to non-irrigated conditions would be required to meet full crop water demand (Fig. 5c). Improving crop water management, by applying farming practices that reduce soil evaporation, retaining more rainwater in the soil, growing more water-saving crops, or applying deficit irrigation[41] can reduce

irrigation water requirements and consequently energy demand for water pumping[47,48].

## Discussion

Our research focuses on the neglected connection between the fossil energy requirements of intensive agriculture and a fundamental component of the global agrarian transition, the one produced by LSLAs. With particular reference to fertilizer usage and irrigation in low- and middle-income countries, we find that a transition to high-input agriculture produced by LSLAs would require 5 times more fossil energy than low-input agriculture. The higher energy-intensity of high-input agriculture over the considered LSLAs (4.07 million hectares or ~0.27% of global croplands extent) would translate into 15 million barrel of oil equivalent per year (~0.04% of global annual oil consumption) and increase GHG emissions by 6 million tons of $CO_2$ per year (or 0.04% of global total GHG

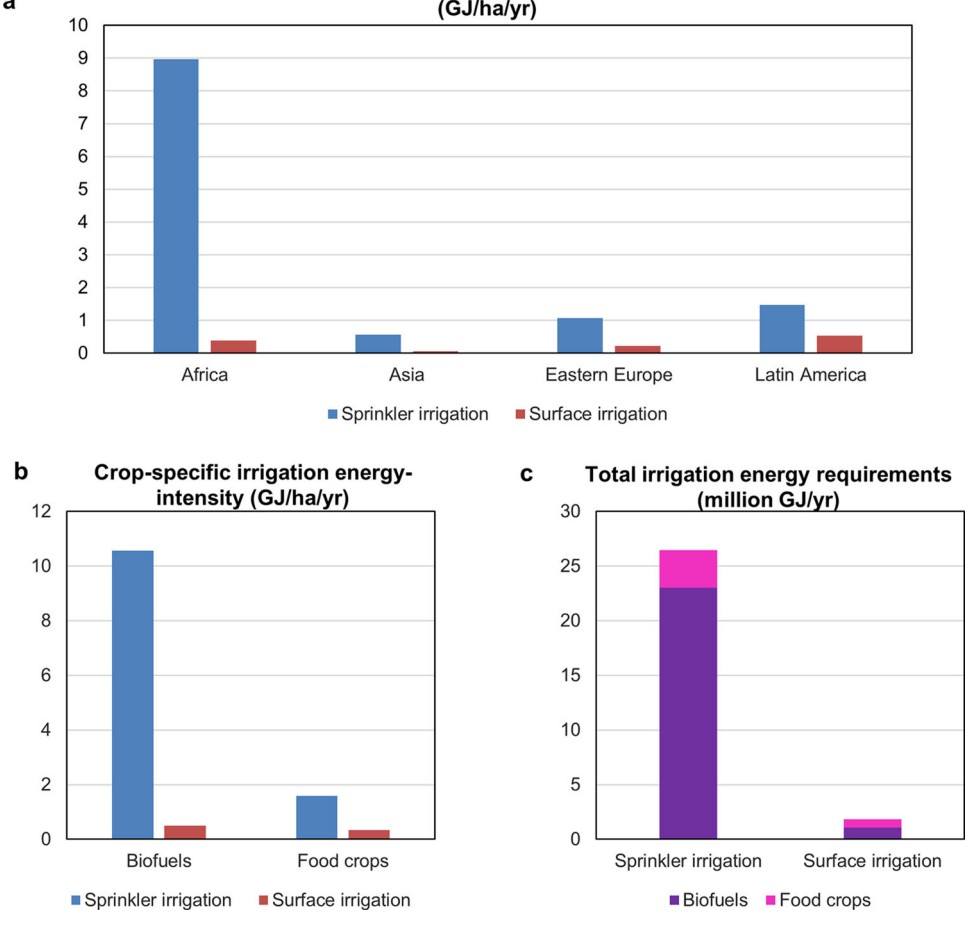

**Fig. 5 Energy demand from irrigation over large-scale land acquisitions (LSLAs). a, b** Average irrigation energy-intensity per region and per crop, respectively. **c** Aggregated irrigation energy requirements (million GJ yr$^{-1}$) for the intended crop area to biofuels and food crop plantations. Irrigation scenarios consider one with sprinkler irrigation systems and one with surface irrigation systems.

**Table 1 Comparison of energy- and greenhouse gas (GHG)-intensities over large-scale land acquisitions (LSLAs).**

| Activity | Fossil-fuel-based energy intensity (GJ ha$^{-1}$) | GHG intensity (tons CO$_2$-eq yr$^{-1}$ ha$^{-1}$) | GHG emission over 197 considered LSLAs (Mtons CO$_2$-eq yr$^{-1}$) | Reference |
|---|---|---|---|---|
| Low-input | 4.1 | 0.3 | 1.5 | This study |
| High-input | 19.1 | 1.6 | 7.2 | This study |
| N$_2$O emissions from high-input synthetic fertilizers | – | 0.3 | 1.3 | This study |
| Irrigation | 3.2 | 0.3 | 2.1 | This study |
| Land use change | – | 14.1 | – | Ref. [28] |

Low-input and high-input show fossil-fuel-based energy use from labor, machinery, fertilizers, chemicals, fuels, and seeds. Land use change shows GHG intensity from deforestation and loss of soil carbon[28]. N$_2$O emissions show GHG emission intensity from synthetic N fertilizer application in high-input farming.

emissions from agriculture). While here we assessed agricultural energy-related GHG emissions needed to close crop-yield gaps, non-energy-related activities, such as land use change, are estimated to have a GHG emission intensity of 14.1 tons CO$_2$-equivalent yr$^{-1}$ ha$^{-1}$ and therefore might be the main drivers of GHG emissions over LSLAs (Table 1).

The global agrarian transition is a broader global phenomenon that far exceeds the footprint of LSLAs considered in this study. Driven by economies of scale, the ongoing industrialization of agriculture is promoting a transition from smallholder to large-scale commercial farming in low- and middle-income countries. Globally, small farms (under 2 hectares) produce 30% of food supply over 24% of agricultural areas (~360 million hectares)[49]

and have an important role in supporting food security of local populations[50]. Assuming that 360 million hectares of global small farms will make a transition from low-input to high-input farming, an additional 0.6 gigatons CO$_2$-equivalent per year would be emitted, or 4% of current global total GHG emissions from agriculture[31].

The Haber-Bosch process to manufacture synthetic fertilizers is estimated to use ~1.2% of energy production and carbon emission worldwide[51]. Many of the major LSLAs will utilize ammonia produced from the energy-intensive Haber-Bosch process. Moreover, after farmers apply synthetic fertilizers to crops, chemical reactions lead to the formation and emission of nitrous oxide, a potent GHG with 265 times more global warming

potential than carbon dioxide[52,53]. Over the considered LSLAs, we estimate that synthetic fertilizer application in high-input farming would release 1.3 million tons $CO_2$-equivalent per year, with a GHG intensity of 0.3 tons $CO_2$-equivalent $yr^{-1} ha^{-1}$ (Table 1). However, less carbon-intensive options to industrial fertilizers are available, such as using recycled organic matter, nutrients recovered from wastewater-treatment plants or manure, a more efficient application of fertilizers[53], or the deployment of cover crops like soy and other legumes that fix atmospheric nitrogen. Even though these less carbon-intensive agroecological alternatives to industrial fertilizers are available, they might be technically and economically difficult to adopt over large farms. The production of fossil-fuel-free ammonia to supplement fertilizers derived from organic materials[54] is also an option to lower the carbon emissions in industrial fertilizer production[55,56].

In terms of the energy associated with irrigation, it is clear that LSLAs are often established on land where the closure of the yield gap of specific crops for commercial agriculture would require irrigation in addition to fertilizer applications. Except for the cases of land suitable for gravity-based irrigation, the withdrawal and delivery of irrigation water from streams, aquifers, or other water bodies to cultivated land typically requires substantial amounts of energy. While energy for fertilizer usage may be employed in supply chains distant from land acquisition, the fuel needed for water delivery must be sourced from the local agrarian economy, where it might compete with demand from other livelihood uses.

The LSLA component of the global agrarian transition has clear implications on multiple dimensions and scales. Focusing on the energy side of transnational land investments provides analytical insight into a series of interdependencies and tradeoffs that emerge from the integrated perspective of the energy–water–food nexus. For example, synergies and tradeoffs between land use for food crops, water use, and energy infrastructure should be considered. The additional energy required by a transition to high-input farming might have direct impact on energy poverty and fairness in access to local energy resources by marginalized rural communities, especially for those communities relying on the collection of local scarce natural resources for energy production[57]. These processes of agrarian transformation can be highly energy-intensive and should be evaluated to avoid further competition with scarce local energy resources. It is fundamental to apply a nexus approach when assessing the sustainability and distribution of benefits of rural transitions, and to understand who are the winners and losers in such transformations. For instance, under suitable environmental and social conditions, small-scale energy infrastructures such as renewable mini-grid[58] can be used by local farmers to access water through, for instance, solar water pumps, thereby allowing for improved energy security with limited land competition with food crops, and at the same time reduce the use of fossil fuel for agricultural production and water use[59]. In this context, the development of small renewable-energy infrastructures at the community level for agricultural production, including energy infrastructure for irrigation purposes, has been proven to be a viable solution to support the livelihood of smallholder farmers, mitigate climate change, reduce energy poverty, and improve local energy access, with positive effects on social justice, such as better distribution of benefits and improved gender equality[60]. For example, small-scale and community agrivoltaics—combining agriculture with solar photovoltaics—can increase crop productivity, while at the same time generating renewable energy for local populations[61].

The energy footprint of LSLAs and their implication for climate change have not received sufficient attention in the scholarly literature and in public assessments of their potential benefits and perils[13,15], nor are they considered in national policies promoting large-scale agricultural investments or in specific land-deal decisions. Specifically, LSLAs have impacts on the energy security of local communities which have not been carefully addressed and which should be considered. Given the high energy demand of fertilizer production and the associated GHG emissions, there is a plausible case for more integrated monitoring of the changes in embedded energy resulting from crop choices and LSLAs. In order to evaluate future directions and options of land-use policy and rural development, the energy-budgeting approach should be integrated to give a more comprehensive account of the implications and tradeoffs, and inform decisions for longer term sustainability.

The speed with which LSLAs and agrarian transitions are taking shape globally may have profound and lasting consequences. While LSLAs might intensify agricultural production, achieve economies of scale, provide infrastructures, and increase agricultural productivity for specific crops, its impacts on socio-environmental sustainability and the distribution of potential benefits between agricultural corporations and local farmers remains highly questionable[16]. There is consolidated knowledge of the long-term unsustainability of energy-intensive agricultural practices because of the massive reliance on fossil-fuel-based energy. Especially for non-industrialized countries, fossil-fuel-based agricultural transitions have been usually funded through public subsidies, and historically suffered from high vulnerability because of their sensitivity to energy crises and volatility of fossil-fuel prices[62,63]. Lack of energy security is still a structural condition in large parts of the rural areas in low- and middle-income countries, where there is little and inconsistent access to affordable and reliable energy for agricultural production[64]. Both fertilizer and irrigation energy usage should be evaluated for transnational and localized impacts, respectively, and LSLA expansion needs to be placed under scrutiny following an integrated food–energy–water approach that prioritizes issues of local resource access and justice. The energy variable in this conversation is therefore not only limited to issues of fossil fuel consumption and its implications for climate change, but it also directly relates to issues of energy access and justice, the food–water–energy nexus, livelihood choices, property rights, and quality of life in the context of smallholder farmers.

## Methods
In this paper, we consider the energy investment of moving from a range of existing land-uses to the intended uses of LSLAs. Using estimates of annual energy input per area of crops from a range of sources (Fig. 3 and Supplementary Table 1), we compare the energy investment for some of the largest crop plantations being proposed for LSLAs based on the Land Matrix dataset[36]. We use a sample of 197 land deals from the Land Matrix dataset[36] and covering an area of about 4.07 million hectares across 39 countries (Fig. 1). The Land Matrix is a joint international initiative collecting data on transnational land acquisitions since 2000. This dataset reports only land investments with areas greater than 200 ha. Following Müller et al.[14], we selected land deals: (1) having their status updated to "contracted", "in start-up phase", or "in production"; (2) intended for agricultural use; (3) for which the coordinates of the deal location were reported in the Land Matrix database.

**Energy usage of low- and high-input agriculture.** To our knowledge, this is the first systematic evaluation of the energy requirements of agricultural intensification through LSLAs. We use large-scale agricultural land-deals as a case study of the energy implication of a new green revolution across LSLAs in low- and middle-income countries. We estimated nitrogen application rates and yield gap-closure levels before LSLAs using the seminal work of Mueller et al.[35]. We consider low-input versus high-input agricultural practices and use the seminal work from Pimentel and Pimentel[34], as well as our new large-scale estimates of energy consumption by irrigation to draw energy comparison estimates (Supplementary Materials). Energy inputs are at the farm level and consider fossil-fuel-based energy from labor, machinery, fertilizers, chemicals, fuels, and seeds. Irrigation water energy inputs were assessed considering irrigation water demand volume[46] and energy requirements for pumping (Supplementary Materials). There are, of course, practical limitations to energy calculations in life cycle assessment methods[65]. For oil palm and jatropha we also show the fossil-fuel-based energy required in the mill

to produce biodiesel (Fig. 3). We then calculate the aggregate energy investment for the crops proposed by current major LSLA investments globally (Figs. 4 and 5) in terms of their energy investment and present potential shifts that could be considered from low-input to high-input agricultural land-use practices.

**Assessment of GHG emissions**. GHG emissions from high-input LSLAs were assessed considering emissions from oil combustion and upstream emissions during crude oil production. According to the US Environmental Protection Agency, average emissions during oil combustion are 429.61 kg $CO_2$ per 42-gallon barrel of oil[66]. Average global emissions from upstream crude oil production processes are 63.03 kg $CO_2$ per 42-gallon barrel of oil emitted in upstream crude oil[67]. Therefore, we assume a GHG emission intensity of oil equal to 492.6 kg $CO_2$ equivalent per barrel of oil. GHG emissions from $N_2O$ from synthetic nitrogen application in high-input farming were assessed considering the Food and Agriculture Organization emissions factor of 0.0132 kg $N_2O$ per kg synthetic N, and assuming a global warming potential from $N_2O$ equal to 265. Nitrogen application rates (kg N per hectare) were taken from the corresponding data source used to determine energy intensities of high-input farming (See Supplementary Table 1).

## Data availability
Data used to support findings of the study are available in the reference list and Supplementary Materials.

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

## Acknowledgements

Publication was made possible in part by support from the Berkeley Research Impact Initative (BRII), sponsored by the UC Berkeley Library. This work was partially supported by the National Socio-Environmental Synthesis Center (SESYNC) under funding received from the National Science Foundation DBI-1639145. J.D.A., M.C.R., and P.D.O. acknowledge support from the European Union's Horizon 2020 research and innovation programme under the Marie Skłodowska-Curie Action (MSCA) Innovative Training Network (ITN) grant agreement No. 861509 - NEWAVE. A.S. acknowledges funding from a Beatriu de Pinós grant, Government of Catalonia's Secretariat for Universities and Research of the Ministry of Economy and Knowledge (2017 BP 00023).

## Author contributions

L.R., M.C.R., S.A., D.D.C., J.D.'A., N.D.M., A.S., G.S., and P.D.'O. conceived, designed, and wrote the study; L.R., M.C.R., S.A., D.D.C., and N.M. performed the analyses; L.R. and S.A. analyzed the data.

## Competing interests

The authors declare no competing interests.
