## [Peer Review File. · Nature Communications]

REVIEWER COMMENTS

Reviewer #1 (Remarks to the Author):

The paper addresses the timely issue of energy-use effects as a consequence of large-scale land acquisitions (LSLAs), and includes a novel energy analysis for the most common irrigation methods. I have some specific suggestions and questions:

Even when the focus is clearly on energy, I think conclusions should pay more attention to other elements, in particular whether "...the withdrawal and delivery of irrigation water from streams, aquifers or other water bodies to cultivated land" is sustainable or feasible at all!

The text does a good job in describing the energy aspects of the investigation but needs a brief overview of the land acquisitions that are now referred to in a single sentence (lines 90-91). Think of readers who had never heard of the Land Matrix Database. What is the time span of those acquisitions? Were they the consequence of a slow, non-aggressive tendency towards larger farms because of economies of scale, or could be described as land grabbing? The area covered (ca. 4 million hectares) needs to be mentioned here, and also the rationale for choosing units of at least 200 ha (line 299).

Industry sources are OK, but should be complemented at least by one additional independent source (lines 158-164). The second sentence of the paragraph can be deleted.

Options to industrial fertilizers in lines 214-215 are not feasible, or become too costly, as the size of the plot and farm goes up. Same, perhaps, for "agroecological practices" (L. 255). Connected to lines 266-68?

L. 258: "maintaining compatibility with small-farmer land tenure..." This seems to go against the whole logic of LSLAs. Please, clarify.

L. 312-13: "...we have focused on the dominant variables of fertilizer and water usage..." Some previous analyses have shown the importance of fuels and machinery.

MINOR

L. 82-83: Aren't most of those emissions caused by livestock? Your paper refers only to crops. Figure 1: Legend does not need "Histograms showing the..."; can start by "Nitrogen application rates..."

L. 125-6: "...their energy intensity is relatively LOW compared to other staple crops."

Figure 3a. : Trees should be 'other trees', since eucalypts (and coffee and rubber trees) have their own category.

L. : 150-7: Number of oil barrels and tons of CO2 are fine and correct, but readers need to put these additional emissions in perspective regarding actual emissions. The reference to specific countries is a good idea. Related to this: Line 53: abstract refers to the "high carbon footprint of LSLAs", but this does not seem to be quantified neither in the Introduction o Discussion sections.

Lines 219-222: Are these geoengineering options?

L. 231: "...usage for irrigation is less than FOR fertilizer production..."

I suggest you consider these two citations:

Hamant, O., 2020. Plant scientists can't ignore Jevons paradox anymore. *Nature Plants*, 6(7), pp.720-722. [regarding oil palm]

Samberg, L.H., Gerber, J.S., Ramankutty, N., Herrero, M. and West, P.C., 2016. Subnational distribution of average farm size and smallholder contributions to global food production.

Reviewer #2 (Remarks to the Author):

The manuscript examines the energy use implications of transitions to higher intensity agricultural systems following large scale land acquisitions (LSLAs). The authors find that LSLAs, which collectively occupy 4.7 million hectares, could result in a 5-fold increase in fossil-based energy use, or increase in GHG emissions of 6 million tonnes of CO₂. The authors briefly discuss other implications of LSLAs, including: increased competition for access to energy and resultant energy poverty and associated inequities; and how LSLAs could also increase water use.

The energy implications, and more broadly speaking the GHG implications, of LSLAs and other agricultural transitions have not been investigated in depth. Given that many existing policy mechanisms support LSLAs and other pathways to agricultural intensification, the analyses in this paper provides a first examination of the fossil energy and associated CO₂ implications of agricultural transitions following LSLAs.

However, results from the analysis indicate that the fossil energy and associated GHG implications of LSLAs are not important in the scope of global energy use or food system emissions. Specifically, the analysis finds the fossil energy implication of LSLAs to be ~15 million barrels of oil equivalent per year, compared to a current global use of ~100 million barrels of oil equivalent per day, while the GHG implications of FLSAs are estimated to be 6 million tons CO₂ per year, compared to current global food system emissions of ~10-15 billion tons CO₂e per year.

I highly recommend reframing the analysis to increase emphasis on some combination of (a) the system wide GHG implications of LSLAs (e.g. what feedback is there between increased fertiliser inputs and increased GHG emissions from N₂O?), (b) other environmental implications of LSLAs, (c) the potential food security implications of LSLAs, and/or (d) the energy competition and inequity implications of LSLAs. (These are listed in no particular order.)

In the context of the analysis on LSLAs and their fossil energy implications, I am concerned that (1) there is limited discussion of the broader energy and agricultural context, and (2) that there is limited/no justification on many of the assumptions in the analysis and the lack of sensitivity analyses around these assumptions.

More on these can be found immediately below:

(1) The broader energy and agricultural context

The paper focuses energy use in LSLAs (rightfully so), but in many cases does so without accounting for the broader context. A few questions on the broader agricultural context are below:

(a) What do knock-on effects of a transition to more intense systems following LSLAs mean for (1) total GHG emissions from that system, and (2) GHG emissions per unit of food produced from that system (as well as fossil energy emissions per unit of food from those systems)? Non-energy GHGs are also important to consider because GHGs from energy use are not the only GHG emissions (and in many cases a small proportion of total GHG emissions) in food systems. This needs to be discussed, or alternatively formally incorporated into the analysis.

(b) Would LSLAs reduce agricultural land, and thus energy and CO₂, elsewhere? The authors mention increasing yields benefits global land use (L72), which seems to imply it would reduce land use elsewhere (and associated fossil energy use and CO₂ emissions). Would it be possible to account for this knock-on implication of LSLAs into the analysis?

(c) How do the estimated implications of LSLAs relate to current global fossil energy use and food

system GHG emissions?

(2) Sensitivity analyses

Many of the assumptions used to examine the fossil energy and GHG impact of LSLAs are reasonable. However, at the same time, the assumptions in the analysis are only one of several potential reasonable assumptions that could be used to examine the fossil energy and GHG impact of LSLAs. Please incorporate some sensitivity analyses to test the robustness of your assumptions and associated results, and to show how e.g. different energy sourcing or different levels of intensity could influence energy transitions stemming from LSLAs. A short list of potential analyses is below:

(a) The authors assume that all LSLAs will transition to high intense farming systems. Is there rationale and justification for this assumption? If there is, please mention the justification for this assumption in the manuscript. If not, please incorporate sensitivity on if e.g. only 75% (and/or other thresholds) of LSLAs transition to high intense systems.

(b) Please add some sensitivity around your assumption that all energy in modern agricultural systems is sourced from oil (from L153-154). One alternative is to use current breakdowns of energy use at a national (or regional) basis, or to assume different combinations of sourcing from e.g. oil and natural gas, etc.

(c) Please provide justification that all LSLAs close yield gaps given that there isn't strong evidence for producers to reduce the gap below .2 (see Lobell et al 2009, Annual Review of Environment and Resources). This is also another location for potential sensitivity, e.g. the energy and CO2 implications if gaps are perfectly closed, closed to a ratio of .1, to a ratio of .2, etc.

In addition, a few in-line comments:

In general: What is the geographic coverage of your LSLA data? L185-195 seem to imply it is global in coverage, but this is never explicitly mentioned.

Abstract: Please incorporate specificity when possible. For instance, the 5-fold increase in energy use translates to 6 million tonnes of CO2.

L104-106: What about differences in management techniques, such as e.g. access to fertiliser, timing of nutrient applications, etc? These are discussed briefly later in the manuscript, but is it possible to incorporate these into the analysis?

L145-146: This assumes a scenario where all LSLAs transition to high intensity and are fueled entirely through oil. The range of scenarios that account for different pathways of development also need to be discussed.

L155-157: Please specify the time frame over the 6 million tons CO2 mentioned here. I think this is per year?

L156 and L207: The stated increase in fossil CO2 emissions resulting from LSLAs in these lines are inconsistent. L156 mentions 6 million tons CO2, whereas L207 mentions 15 million tons. I think these should both be 6 million tons following the math in L145-157?

L223-236: Do you have evidence for this local competition of energy resources? This is really important in the context of LSLAs (and other aspects of development), but there are currently no references in this section that support the chain of LSLAs -> more energy use -> increased local competition -> energy poverty, inequity, and other knock-on impacts. You've articulated the first step (LSLAs -> more energy use) in the manuscript, but have not provided evidence for the other two.

L298-300: Does your analysis include all land deals > 200 hectares from Land Matrix? Or just a subset of them? If it is a subset, why have you used only a subset? From Land Matrix, it appears that there are >2000 (based on the land deal size) to ~770 (based on land in current operation) LSLAs that occupy > 200 hectares, so it is unclear how or why the subset of 207 LSLAs used in the analysis was decided upon.

Figure 3a: This is a very clear visualization of your LSLA data. Could you have a similar visualization that shows the geographical distribution of LSLAs?

REVIEWER COMMENTS

Reviewer #1 (Remarks to the Author):

The paper addresses the timely issue of energy-use effects as a consequence of large-scale land acquisitions (LSLAs), and includes a novel energy analysis for the most common irrigation methods. I have some specific suggestions and questions:

Thank you for the constructive and thorough review.

Even when the focus is clearly on energy, I think conclusions should pay more attention to other elements, in particular whether "...the withdrawal and delivery of irrigation water from streams, aquifers or other water bodies to cultivated land" is sustainable or feasible at all!

Our previous work has mapped areas of the world where the local water resources are not sufficient to meet the local irrigation water requirements. In those regions irrigation would occur at the expenses of environmental flows and/or groundwater stocks (i.e., irrigation is "unsustainable" from the standpoint of water resource use), should farmers decide to irrigate the land. Capitalizing on those results, we have evaluated to what extent LSLAs are taking place in areas unsuitable for sustainable irrigation:

"In addition to nutrients, water is another important input necessary to increase yields and avoid crop growth under water stress³⁹. The use of irrigation enables reliable water supply while boosting crops productivity⁴⁰. However, in many geographical settings, irrigation practices are unsustainable because their water requirements exceed local renewable water availability and irrigation induces the depletion of environmental flows and groundwater stocks⁴¹. Using recent irrigation sustainability assessments⁴², we find that 20% of land deals were located in regions where water resources are insufficient to sustainably meet irrigation demand."

The text does a good job in describing the energy aspects of the investigation but needs a brief overview of the land acquisitions that are now referred to in a single sentence (lines 90-91). Think of readers who had never heard of the Land Matrix Database. What is the time span of those acquisitions? Were they the consequence of a slow, non-aggressive tendency towards larger farms because of economies of scale, or could be described as land grabbing?

Thank you, we have added to the introduction some further explanation of and references to LSLAs and the Land Matrix. Specifically, we provide in the revised version a definition, clarify the timeframe of LSLAs and their context, and added literature on their role for agricultural development. This section in the introduction now reads:"

Controversies over the role of agricultural intensification for land use and users have intensified with the recent unprecedented rise in large-scale land acquisitions (LSLA) globally^{11,12}. LSLAs refer to long-term and large-scale acquisitions of land property or use

rights through domestic and foreign actors, which have been sparked by – among other factors – food security concerns and the rediscovery of agriculture as a key investment sector following the 2008 food, financial and energy crises¹³. According to the Land Matrix – a joint international initiative collecting data on LSLAs since 2000 – 90 million hectares of land (about the surface area of Venezuela) have been acquired globally by investors since 2000 (ref. 14).

A few studies have discussed the potential opportunities of LSLAs for agricultural development, for instance, their potential to increase yields¹³ and economic benefits for the country through large-scale investments into farmland¹⁵. However, other studies have described these land deals as “land grabs” that entail a vast range of adverse impacts on local users, including land dispossession and livelihood loss¹⁶⁻¹⁸. There is strong evidence that large-scale investments in agriculture by agribusiness corporations have been implemented at the cost of small-holders, traditional users, Indigenous people and more vulnerable segments of the rural population^{12,16}. LSLAs could paradoxically increase crop production, while undermining local food security through the production of energy-rich but nutrient-poor crops for export markets¹⁴. While this radical transformation of agrarian systems through LSLAs has been extensively investigated in relation its political implications^{19,20} and its impacts on property systems²¹, rural livelihoods²², crop yields¹⁴, water use and redistribution²³, food security²⁴⁻²⁶, environmental impacts²⁷, and carbon emissions²⁸. However, with few exceptions²⁹, its energy implications remain poorly understood.”

Furthermore, in the revised manuscript, we added a new figure (Figure 1) showing the geographic distribution of the LSLAs considered in this study.

Figure 1. Geographical distribution of LSLAs considered in this study. We consider 197 land deals for agricultural use for which the geographic coordinates were available in the Land Matrix database³⁴. These land deals are located in 39 countries and account for 4.07 million hectares of acquired land across Africa as (73 deals, 2.4 Mha), Asia (43 deals and 0.58 Mha), Europe (33 deals and 0.54 Mha), and Latin America (11 deals and 0.55 Mha).

The area covered (ca. 4 million hectares) needs to be mentioned here, and also the rationale for choosing units of at least 200 ha (line 299).

The text now reads:

We consider a sample of 197 lands deals from the Land Matrix dataset³⁴ and covering an area of about 4.07 million hectares across 39 countries (Figure 1). The Land Matrix is a joint international initiative collecting data on transnational land acquisitions since 2000. This datasets reports only land investments with areas greater than 200 ha. Following Müller et al., 2021 (ref. 14), we selected land deals: 1) having their status updated to “contracted”, “in start-up phase”, or “in production”; 2) intended for agricultural use; 3) for which the coordinates of the deal location were reported in the Land Matrix database.”

Options to industrial fertilizers in lines 214-215 are not feasible, or become too costly, as the size of the plot and farm goes up. Same, perhaps, for “agroecological practices” (L. 255). Connected to lines 266-68?

We agree with this comment, the text in the discussion section now reads:

“ Even though these agroecological less carbon-intensive alternatives to industrial fertilizers are available, they might be technically and economically difficult to adopt over large farms.”

L. 258: “maintaining compatibility with small-farmer land tenure...” This seems to go against the whole logic of LSLAs. Please, clarify.

We agree. This phrase was deleted.

MINOR

L. 82-83: Aren't most of those emissions caused by livestock? Your paper refers only to crops.

We revised this section and the text now reads:

“Global food systems are major energy users and contributors to climate change. Food systems consume 15-30% of global primary energy³⁰ and emit 25% of global total greenhouse gas (GHG) emissions (about 14.6 gigatons CO₂-equivalent in 2017)³¹. Although 35% of these emissions are caused by dairy and meat production, the remaining share is from activities pertaining to crop production for direct human consumption³¹. While, GHG emissions associated with deforestation and land use change over 40 million hectares of LSLAs have been recently estimated to 8 gigatons CO₂-equivalent in the 2000-2016 period²⁸, the energy and related GHG emissions associated with the agrarian transition induced by LSLAs and other dynamics favoring the expansion of commercial farming have often been overlooked.”

Figure 1: Legend does not need “Histograms showing the...”; can start by “Nitrogen application rates...”

Done, thank you

L. 125-6: “...their energy intensity is relatively LOW compared to other staple crops.”

Done, thank you

Figure 3a. : Trees should be 'other trees', since eucalypts (and coffee and rubber trees) have their own category.

Done, thank you

L. : 150-7: Number of oil barrels and tons of CO2 are fine and correct, but readers need to put these additional emissions in perspective regarding actual emissions. The reference to specific countries is a good idea.

Thank you

Line 53: abstract refers to the "high carbon footprint of LSLAs", but this does not seem to be quantified neither in the Introduction or Discussion sections.

In the revised manuscript we also assessed carbon footprints.

L. 231: "...usage for irrigation is less than FOR fertilizer production..."

Done, thank you

I suggest you consider these two citations:

Hamant, O., 2020. Plant scientists can't ignore Jevons paradox anymore. *Nature Plants*, 6(7), pp.720-722. [regarding oil palm]

Samberg, L.H., Gerber, J.S., Ramankutty, N., Herrero, M. and West, P.C., 2016. Subnational distribution of average farm size and smallholder contributions to global food production. *Environmental Research Letters*, 11(12), p.124010.

Thank you for these suggestions. We have now cited these 2 studies in the text by expanding the introduction and discussion sections.

Reviewer #2 (Remarks to the Author):

The manuscript examines the energy use implications of transitions to higher intensity agricultural systems following large scale land acquisitions (LSLAs). The authors find that LSLAs, which collectively occupy 4.07 million hectares, could result in a 5-fold increase in fossil-based energy use, or increase in GHG emissions of 6 million tonnes of CO₂. The authors briefly discuss other implications of LSLAs, including: increased competition for access to energy and resultant energy poverty and associated inequities; and how LSLAs could also increase water use.

The energy implications, and more broadly speaking the GHG implications, of LSLAs and other agricultural transitions have not been investigated in depth. Given that many existing policy mechanisms support LSLAs and other pathways to agricultural intensification, the analyses in this paper provides a first examination of the fossil energy and associated CO₂ implications of agricultural transitions following LSLAs.

Thank you for the constructive and thorough review.

However, results from the analysis indicate that the fossil energy and associated GHG implications of LSLAs are not important in the scope of global energy use or food system emissions. Specifically, the analysis finds the fossil energy implication of LSLAs to be ~15 million barrels of oil equivalent per year, compared to a current global use of ~100 million barrels of oil equivalent per day, while the GHG implications of LSLAs are estimated to be 6 million tons CO₂ per year, compared to current global food system emissions of ~10-15 billion tons CO₂e per year.

I highly recommend reframing the analysis to increase emphasis on some combination of (a) the system wide GHG implications of LSLAs (e.g. what feedback is there between increased fertilizer inputs and increased GHG emissions from N₂O?),

The focus of this study is not to determine GHG emissions from LSLAs, but rather to estimate the energy costs of agricultural inputs that are needed to close yield gaps. However, we agree with the reviewer that an additional analysis on GHG emissions from N₂O releases due to synthetic fertilizers would improve the study. Therefore, we estimated GHG emissions from N₂O emissions due to high-input fertilizers adoption.

The text now reads:"

Over considered LSLAs, we estimate that synthetic fertilizers application in high input farming would release 1.3 million tons CO₂-equivalent per year, with a GHG intensity of 0.3 tons CO₂-eq yr⁻¹ ha⁻¹ (Table 1)."

(b) other environmental implications of LSLAs,

(c) the potential food security implications of LSLAs, and/or

We have expanded the introduction section covering these suggestion, which now reads:” *Controversies over the role of agricultural intensification for land use and users have intensified with the recent unprecedented rise in large-scale land acquisitions (LSLA) globally^{11,12}. LSLAs refer to long-term and large-scale acquisitions of land property or use rights through domestic and foreign actors, which have been sparked by – among other factors – food security concerns and the rediscovery of agriculture as a key investment sector following the 2008 food, financial and energy crises¹³. According to the Land Matrix – a joint international initiative collecting data on LSLAs since 2000 – 90 million hectares of land (about the surface area of Venezuela) have been acquired globally by investors since 2000 (ref. 14).*

A few studies have discussed the potential opportunities of LSLAs for agricultural development, for instance, their potential to increase yields¹³ and economic benefits for the country through large-scale investments into farmland¹⁵. However, other studies have described these land deals as “land grabs” that entail a vast range of adverse impacts on local users, including land dispossession and livelihood loss¹⁶⁻¹⁸. There is strong evidence that large-scale investments in agriculture by agribusiness corporations have been implemented at the cost of small-holders, traditional users, Indigenous people and more vulnerable segments of the rural population^{12,16}. LSLAs could paradoxically increase crop production, while undermining local food security through the production of energy-rich but nutrient-poor crops for export markets¹⁴. While this radical transformation of agrarian systems through LSLAs has been extensively investigated in relation its political implications^{19,20} and its impacts on property systems²¹, rural livelihoods²², crop yields¹⁴, water use and redistribution²³, food security²⁴⁻²⁶, environmental impacts²⁷, and carbon emissions²⁸. However, with few exceptions²⁹, its energy implications remain poorly understood.”

(d) the energy competition and inequity implications of LSLAs. (These are listed in no particular order.)

The text now reads:”

The LSLAs component of the global agrarian transition has clear implications on multiple dimensions and scales. Focusing on the energy side of transnational land investments provides analytical insight into a series of interdependencies and tradeoffs that emerge from the integrated perspective of the energy-water-food nexus. For example, synergies and tradeoffs between land use for food crops, water use and energy infrastructure should be considered. The additional energy required by a transition to high-input farming might have direct impact on energy poverty and fairness in access to local energy resources by marginalized rural communities, especially for those communities relying on the collection of local scarce natural resources for energy production⁵⁵. These processes of agrarian transformation can be highly energy intensive and should be evaluated to avoid further competition with scarce local energy resources. It is fundamental to apply a nexus approach when assessing the sustainability and distribution of benefits of rural transitions and to understand who are the winners and losers in such transformations. For instance, under suitable environmental and social conditions, small-scale energy infrastructures such as renewable mini-grid⁵⁶ can be used by local farmers to access water through for instance solar

water pumps, thereby allowing for improved energy security with limited land competition with food crops and at the same time reduce the use of fossil fuel for agricultural production and water use⁵⁷. In this context, the development of small renewable energy infrastructures at the community level for agricultural production, including energy infrastructure for irrigation purposes, has been proven to be a viable solution to support the livelihood of smallholder farmers, mitigate climate change, reduce energy poverty, and improve local energy access, with positive effects on social justice, such as better distribution of benefits and improved gender equality⁵⁸. For example, small-scale and community agrivoltaics – combining agriculture with solar photovoltaics – can increase crop productivity, while at the same time generating renewable energy for local populations⁵⁹.”

In the context of the analysis on LSLAs and their fossil energy implications, I am concerned that (1) there is limited discussion of the broader energy and agricultural context, and (2) that there is limited/no justification on many of the assumptions in the analysis and the lack of sensitivity analyses around these assumptions.

More on these can be found immediately below:

(1) The broader energy and agricultural context

The paper focuses energy use in LSLAs (rightfully so), but in many cases does so without accounting for the broader context. A few questions on the broader agricultural context are below:

(a) What do knock-on effects of a transition to more intense systems following LSLAs mean for (1) total GHG emissions from that system, and (2) GHG emissions per unit of food produced from that system (as well as fossil energy emissions per unit of food from those systems)? Non-energy GHGs are also important to consider because GHGs from energy use are not the only GHG emissions (and in many cases a small proportion of total GHG emissions) in food systems. This needs to be discussed, or alternatively formally incorporated into the analysis.

The focus and novelty of this study is not to determine GHG emissions from LSLAs, but rather to estimate the energy costs of agricultural inputs that are needed to close yield gaps. We do agree with the reviewer that GHG emissions from land use change have a larger carbon footprint than energy related emissions in agriculture. Building on a recent study that estimated GHG from land use change over LSLAs globally, the discussion section now reads:”

While here we assessed agricultural energy related GHG emissions needed to close crop yield gaps, non-energy related activities, such as land use change, are estimated to have a GHG emission intensity of 14 tons CO₂-eq yr⁻¹ ha⁻¹ and therefore might be the main drivers of GHG emissions over LSLAs (Table 1).

Table 1. Comparison of energy- and GHG-intensities over LSLAs. Low-input and high-input show fossil-fuel-based energy use from labor, machinery, fertilizers, chemicals, fuels, and seeds. Land use change shows GHG intensity from deforestation and loss of soil carbon

(Liao et al., 2021). N₂O emissions show GHG emission intensity from synthetic N fertilizers application in high-input farming.

Activity	Fossil-based energy intensity (GJ ha ⁻¹)	GHG intensity (tons CO ₂ -eq yr ⁻¹ ha ⁻¹)	GHG emission over 197 considered LSLAs (Mtons CO ₂ -eq yr ⁻¹)	Reference
Low-input	4.1	0.3	1.5	This study
High-input	19.1	1.6	7.2	This study
N ₂ O emissions from high input synthetic fertilizers	-	0.3	1.3	This study
Irrigation	3.2	0.3	2.1	This study
Land use change	-	14.1	-	Liao et al., 2021

“

Moreover, in the introduction section reads:

” *“Global food systems are major energy users and contributors to climate change. Food systems consume 15-30% of global primary energy³⁰ and emit 25% of global total greenhouse gas (GHG) emissions (about 14.6 gigatons CO₂-equivalent in 2017)³¹. Although 35% of these emissions are caused by dairy and meat production, the remaining share is from activities pertaining to crop production for direct human consumption³¹. While, GHG emissions associated with deforestation and land use change over 40 million hectares of LSLAs have been recently estimated to 8 gigatons CO₂-equivalent in the 2000-2016 period²⁸, the energy and related GHG emissions associated with the agrarian transition induced by LSLAs and other dynamics favoring the expansion of commercial farming have often been overlooked.”*

(b) Would LSLAs reduce agricultural land, and thus energy and CO₂, elsewhere? The authors mention increasing yields benefits global land use (L72), which seems to imply it would reduce land use elsewhere (and associated fossil energy use and CO₂ emissions). Would it be possible to account for this knock-on implication of LSLAs into the analysis?

To account for the knock-on effect of LSLAs it would require a computable general equilibrium model of the global agricultural economy, which is beyond the scope of this study. There is a wide body of literature on the rebound effects from increased resource efficiency in agriculture; these processes can limit the benefits of any land sparing due to intensification of LSLAs. We have now added a phrase in the introduction to mention this important point, the text now reads:

” Advances in the nitrogen fertilizer industry, irrigation, and other technologies increased land efficiency in agriculture² but did not necessarily lead to inputs savings³⁻⁵. ”

(c) How do the estimated implications of LSLAs relate to current global fossil energy use and food system GHG emissions?

The discussion section now reads:”

This higher energy-intensity of high input agriculture over considered LSLAs (4.07 million hectares or ~0.27% of global croplands extent) would translate into 15 million barrel of oil equivalent per year (~0.04% of global annual oil consumption) and increase greenhouse gas emissions by 6 million tons of CO₂ per year (or 0.04% of global total greenhouse gas emissions from agriculture). “

(2) Sensitivity analyses

Many of the assumptions used to examine the fossil energy and GHG impact of LSLAs are reasonable. However, at the same time, the assumptions in the analysis are only one of several potential reasonable assumptions that could be used to examine the fossil energy and GHG impact of LSLAs. Please incorporate some sensitivity analyses to test the robustness of your assumptions and associated results, and to show how e.g. different energy sourcing or different levels of intensity could influence energy transitions stemming from LSLAs. A short list of potential analyses is below:

(a) The authors assume that all LSLAs will transition to high intense farming systems. Is there rationale and justification for this assumption? If there is, please mention the justification for this assumption in the manuscript. If not, please incorporate sensitivity on if e.g. only 75% (and/or other thresholds) of LSLAs transition to high intense systems. (b) Please add some sensitivity around your assumption that all energy in modern agricultural systems is sourced from oil (from L153-154). One alternative is to use current breakdowns of energy use at a national (or regional) basis, or to assume different combinations of sourcing from e.g. oil and natural gas, etc. (c) Please provide justification that all LSLAs close yield gaps given that there isn't strong evidence for producers to reduce the gap below .2 (see Lobell et al 2009, Annual Review of Environment and Resources). This is also another location for potential sensitivity, e.g. the energy and CO₂ implications if gaps are perfectly closed, closed to a ratio of .1, to a ratio of .2, etc.

L104-106: What about differences in management techniques, such as e.g. access to fertilizer, timing of nutrient applications, etc? These are discussed briefly later in the manuscript, but is it possible to incorporate these into the analysis?

L145-146: This assumes a scenario where all LSLAs transition to high intensity and are fueled entirely through oil. The range of scenarios that account for different pathways of development also need to be discussed.

To address these points we performed a sensitivity analysis considering a conservative scenario. The results section now reads:”

Under a business as usual strategy, large scale farms are expected to be established on the entire transacted area and are expected to be cultivated and used for intensified crop production²⁸ with high-inputs to maximize agricultural productivity. This implementation strategy yields the highest potential energy use and provides an upper-bound estimate. Similarly, an upper-bound greenhouse gas emissions estimate is provided assuming that additional energy demand is met entirely with fossil-fuels-based energy sources. Different pathways of development can fuel high-input agriculture through less carbon intensive energy sources and therefore reduce carbon emissions. Inclusion of these factors, as well as lower attainable yields, and agroecological practices such as crop diversification, natural fertilizer use, biological pest control, deficit irrigation, and other soft-path water management practices are expected to reduce the amount of fertilizers and irrigation water and therefore reduce energy usage. We tested the sensitivity of our results by assuming that 75% of the transacted deal area is cultivated and considering a 80% attainable yield. Moreover, considering global primary energy demand statistics, we assumed that 80% of energy inputs are powered by fossil fuels and the remaining fraction by renewable energy³⁸. This conservative scenario would lower GHG emissions of high-input farming by 3 million tons of CO₂ per year.”

In addition, a few in-line comments:

In general: What is the geographic coverage of your LSLA data? L185-195 seem to imply it is global in coverage, but this is never explicitly mentioned.

In the revised manuscript we added a map that shows the geographic coverage of considered LSLAs.

Figure 1. Geographical distribution of LSLAs considered in this study. We consider 197 land deals for agricultural use for which the geographic coordinates were available in the Land Matrix database³⁴. These land deals are located in 39 countries and account for 4.07 million hectares of acquired land across Africa as (73 deals, 2.4 Mha), Asia (43 deals and 0.58 Mha), Europe (33 deals and 0.54 Mha), and Latin America (11 deals and 0.55 Mha).

Abstract: Please incorporate specificity when possible. For instance, the 5-fold increase in energy use translates to 6 million tonnes of CO₂.

Done, thank you

L155-157: Please specify the time frame over the 6 million tons CO₂ mentioned here. I think this is per year?

Correct, it is per year.

L156 and L207: The stated increase in fossil CO₂ emissions resulting from LSLAs in these lines are inconsistent. L156 mentions 6 million tons CO₂, whereas L207 mentions 15 million tons. I think these should both be 6 million tons following the math in L145-157?

Thank you for pointing out this typo. It is 6 million tons CO₂.

L223-236: Do you have evidence for this local competition of energy resources? This is really important in the context of LSLAs (and other aspects of development), but there are currently no references in this section that support the chain of LSLAs -> more energy use -> increased local competition -> energy poverty, inequity, and other knock-on impacts. You've articulated the first step (LSLAs -> more energy use) in the manuscript, but have not provided evidence for the other two.

See point d above.

L298-300: Does your analysis include all land deals > 200 hectares from Land Matrix? Or just a subset of them? If it is a subset, why have you used only a subset? From Land Matrix, it appears that there are >2000 (based on the land deal size) to ~770 (based on land in current operation) LSLAs that occupy > 200 hectares, so it is unclear how or why the subset of 197 LSLAs used in the analysis was decided upon.

The methods section now reads:"

We consider a sample of 197 lands deals from the Land Matrix dataset³⁴ and covering an area of about 4.07 million hectares across 39 countries (Figure 1). The Land Matrix is a joint international initiative collecting data on transnational land acquisitions since 2000. This datasets reports only land investments with areas greater than 200 ha. Following Müller et al., 2021 (ref. 14), we selected land deals: 1) having their status updated to "contracted", "in start-up phase", or "in production"; 2) intended for agricultural use; 3) for which the coordinates of the deal location were reported in the Land Matrix database."

Figure 3a: This is a very clear visualization of your LSLA data. Could you have a similar visualization that shows the geographical distribution of LSLAs?

In the revised manuscript, we added a new figure (Figure 1) showing the geography of LSLAs considered in this study.

REVIEWERS' COMMENTS

Reviewer #1 (Remarks to the Author):

I thank the authors for paying such close attention to my concerns and suggestions. Kudos on the inclusion of the new map, which together with the new text summarizes superbly the context and background information I was asking for.

Reviewer #2 (Remarks to the Author):

I appreciate the time and effort the authors spent in revising their manuscript. The manuscript is clear, concise, and easy to follow.

I have two additional very minor comments:

L107: It might be useful to add a range for agriculture-related GHG emissions to illustrate that there's still a bit of uncertainty on this, as you have done for primary energy use. Rosenzweig et al 2020 (Nature Food) could be a helpful citation for this, as would the complementary chapter from the IPCC's Special Report on Land Use and Climate Change.

L218-219: Useful to directly contrast the 3Mt CO₂ mitigation estimate stemming from the uncertainty analysis with the 6Mt CO₂ estimate that stems from the main analysis?

Manuscript # NCOMMS-20-49281A

Reviewer #1 (Remarks to the Author):

I thank the authors for paying such close attention to my concerns and suggestions. Kudos on the inclusion of the new map, which together with the new text summarizes superbly the context and background information I was asking for.

Thank you for your time and constructive review.

Reviewer #2 (Remarks to the Author):

I appreciate the time and effort the authors spent in revising their manuscript. The manuscript is clear, concise, and easy to follow.

Thank you for your time and constructive review.

I have two additional very minor comments:

L107: It might be useful to add a range for agriculture-related GHG emissions to illustrate that there's still a bit of uncertainty on this, as you have done for primary energy use. Rosenzweig et al 2020 (Nature Food) could be a helpful citation for this, as would the complementary chapter from the IPCC's Special Report on Land Use and Climate Change.

Done

L218-219: Useful to directly contrast the 3Mt CO₂ mitigation estimate stemming from the uncertainty analysis with the 6Mt CO₂ estimate that stems from the main analysis?

Done